## OPINION

# What homeostasis leaves out: Kinorhesis, a physiological principle of transformation

**Nelson D. Horseman**

*Physiology Department, University of Cincinnati, Cincinnati, Ohio, USA*

Email: nelson.horseman@uc.edu, n.horseman@amelgo.com

Handling Editors: Kim Barrett & Vaughan Macefield

The peer review history is available in the Supporting Information section of this article (https://doi.org/10.1113/JP291068#support-information-section).

Every physiology textbook and course teaches about homeostasis as a fundamental theory or organizing principle of physiology, and the predominating emphasis in teaching physiology is on the maintenance of a 'stable internal milieu' as originally articulated by Claude Bernard (1885). Despite the extraordinary impact of the theory of homeostasis, there are transformative life-cycle events that cannot be explained by homeostasis. And transformative life-cycle processes, like homeostasis, are ubiquitous in nature. Organisms experience these episodes of transformation as they undergo development and reproduction. These events clearly are physiological even though they are explicitly not homeostatic. Consequently alternative language and concepts are needed to encompass these episodes of biological transformation.

In two recent papers I have proposed a complementary principle termed 'kinorhesis' (from Greek, propel + flow) to account for the physiological processes that control and execute episodes of transformation during development and reproduction (Horseman, 2025a, 2025b). Here I intend to summarize this principle and distinguish this idea from the many critiques of homeostasis that have populated the physiological literature. In doing so I will emphasize that a theory of kinorhesis is not a critique of homeostasis but rather a complementary idea that preserves and reinforces the theory of homeostasis.

Kinorhesis is a physiological principle stating that 'organismal life cycles are conditioned upon episodes of transformation that interrupt intervals of relative stasis so as to bring about developmental and reproductive events'. These physiological processes are predictable even though they are not homeostatic. Kinorhetic processes force the organism through changes in function, metabolism, morphology and behaviour so as to accomplish development and reproduction.

## Discontent about homeostasis

There have been wide-ranging discussions and arguments about exactly what homeostasis is and how it works, and here I list a few papers that I found informative and useful guides to the rest of the literature (Bechtel & Bich, 2024; 2025; Billman, 2020; Dallman, 2003; Davies, 2016; Houk, 1988; McEwen & Wingfield, 2010; Modell et al., 2015; Ramsay & Woods, 2014). Preoccupation with homeostasis is most obviously justified by the critical importance of understanding homeostasis in the practice of medicine, and I will come back to this in a bit. But first if one takes a tour through recent literature that addresses homeostasis from a general or theoretical perspective, the individual would find numerous discussions about how we might need to reframe the concept of homeostasis. These discussions have led to a proliferation of words and phrases that are offered as being more apt than homeostasis, at least in certain situations. Among these are heterostasis, allostasis, homeorhesis, dynamic homeostasis, reactive homeostasis, anticipatory homeostasis, rheostasis and poikilostasis (Selye, 1956; Sterling & Eyer, 1988; Davies, 2016; Bauman & Currie, 1980; McEwen & Wingfield, 2010; Mrosovsky, 1990). Note that all of these terms share at least one of the root syllables *homeo-* or *-stasis*, and this is because they are all concerned with the stabilizing physiological mechanisms addressed by the principle of homeostasis.

The arguments centre on how much variation is too much to be called homeostasis. Is a fever or a circadian body temperature rhythm still homeostatic, or does it need another name? If homeostasis gets disturbed (stressed) over and over again, does it then need a different name? If metabolic demands undergo a big change,

do we need a new word? Though some have tried it has been hard for anyone to umpire these contests, and I am not going to try to do so. I will however offer the observation that despite these discontented offerings, homeostasis is well understood as a general physiological principle. Physicians use their understanding of homeostasis almost constantly to practice medicine, and basic biology teachers are able to convey a fundamental understanding of homeostasis without needing all the varieties of homeostasis-like words. Specialists may continue to find specific language that suits their particular circumstances, but homeostasis as a general principle has been remarkably resilient.

Although some physiologists have fretted over how well the teacups are arranged on the homeostasis table, there has been an elephant in the room whose presence has been tolerated but never explicitly acknowledged. That elephant is the authentically non-homeostatic physiology associated with transformative changes that organisms must undergo for reproduction and development. In my two recent papers I have made a case for recognizing a separate organizing principle of physiology termed 'kinorhesis' to capture the phenomenology and mechanisms of transformative physiological changes (Horseman, 2025a, 2025b). Here is a simple example of what I mean by kinorhesis: a few days ago you were a fish-like thing swimming in a pond, breathing through gills and grazing on algae. Your thyroid gland became activated and set off a sequence of major transformative changes so that now you are hopping about on land taking air via your lungs and eating flies by flinging your tongue out to catch them. Physiology has been doing a whole host of things that are not homeostasis. It has been controlling and executing a programme of profound qualitative transformations that are functional, behavioural, morphological and metabolic. And those changes have occurred at all levels of organization, from subcellular to the whole body. Metamorphosis is an episode of transformation that has many specific facets, only some of which are understood. Episodes of transformation such as metamorphosis occur via physiological processes operating in parallel with homeostasis, but they are not explainable as homeostatic stability.

The Journal of Physiology

There are examples of metamorphosis found across most animal phyla, and metamorphosis is only one example of transformative physiological processes.

The whole domain of physiology that I am calling kinorhesis encompasses the events and processes that are generally associated with reproduction and development. A useful way of thinking about kinorhesis and homeostasis is that homeostasis operates continuously at relatively short time scales to maintain quantitative stability, whereas kinorhesis operates episodically at longer time scales to execute qualitative transformative changes. Homeostasis is necessary for survival of an individual in the present time, and kinorhesis is necessary for completing the programmes of organismal life histories and thereby perpetuating the species. The transformative tasks of physiology are complementary with homeostasis. Homeostasis does not necessarily disappear during episodes of kinorhesis, but without disrupting homeostasis the most interesting aspects of life history – development from a zygote to maturity and reproducing a new generation – could not exist.

Kinorhetic physiology has always lived anonymously within physiology departments, courses and textbooks, mostly in the chapters at the end of the book and the lectures late in the semester. However the transformative changes necessary for reproduction and development have remained outside the explanatory power of homeostasis. Pregnancy illustrates this notion. A pregnant female experiences many homeostatic reflexes and adaptations. Blood pressure, nutrition, renal and immune functions all undergo well-known homeostatic responses during pregnancy (Thornburg, et al., 2006). But none of the homeostatic responses to the pregnancy explains or illuminates pregnancy itself. Pregnancy, *per se*, is clearly not homeostatic. It is a physiological process of profound transformations, some of which reverse after parturition whereas others become permanent parts of the maternal physiology. Understanding pregnancy comprehensively requires considering both the transformative changes and the maintenance of stabilities, and the articulations between these two imperatives.

The initial paper to introduce kinorhesis approaches the topic from a philosophical and theoretical point of view (Horseman, 2025a). Physiology is the study of processes, relationships and interactions that manifest the functions of organisms. These are abstract concepts that are set within physiology's theoretical framework. Theories in physiology fall into three categories. First there are many local theories that provide explanations for function in particular cells and organ systems (Starling's law, Hodgkin and Huxley's formulations, etc.). Secondly there are generalizations from empirical findings that are mostly true, but they are not formal because there are many exceptions (germ theory, law of independent assortment, allometric scaling rules, etc.). Then there are general principles (theories) that apply across levels of organization and phylogenetic lines and provide general umbrella explanations for how organisms are alive. The 'central dogma' of genetic encoding and homeostasis are general principles. And kinorhesis also fits as a general principle. This first paper also describes how biology is served by both physiology theories and evolutionary theories.

A second paper (Horseman, 2025b) approaches kinorhesis from a biological, rather than philosophical, point of view. It reviews a wide range of biological examples from bacteria, plants and animals to illustrate the universality of kinorhesis. Examples are described showing how kinorhesis and homeostasis can coexist benignly but sometimes come into conflict. The paper also delves into kinorhetic regulatory mechanisms and discusses how mechanisms differ between homeostasis and kinorhesis. This paper also discusses the interfaces between physiology and evolution in the context of a theory of kinorhesis. A revived interest in the physiology of evolution has recently been discussed in a special issue series of papers in *The Journal of Physiology* (Noble & Joyner, 2024). Further studies of these implications will be fruitful.

## Homeostasis is priority 1 for medicine because of its acuteness

The prominent place that has been held by homeostasis needs to be understood and appreciated. Modern physiology has its historical roots in human medicine, and medicine is mostly the practical application of homeostasis. Clinical laboratory tests generally ask which physiological variables are within the homeostatic range and whether there are informative relationships between variables that deviate from homeostatic normal. Those relationships can distinguish between simple dehydration

that can be fixed by providing fluids and dehydration caused by diabetes, which will require addressing the underlying hormonal disease. Disturbances to homeostasis can decompensate rapidly if left untreated.

The theory of homeostasis has become a bedrock of not only human physiology but also biology in general. Bernard's original concept was that 'stability of the internal milieu' was a special feature of warm-blooded animals (birds and mammals/humans), allowing them to be highly mobile and to engage in complex behaviours. Before the middle of the 20th century, physiologists focused almost exclusively on mammalian systems (Cannon, 1929; 1939). By the end of World War II, concepts of control systems in animals and their analogues in engineering were formalized by Norbert Weiner and published as *Cybernetics: Or Control and Communication in the Animal and the Machine* (Weiner, 1948). Theoretical treatment of homeostasis and cybernetics makes it possible to describe common control mechanisms across biology and find examples of homeostasis in invertebrates, plants and microbes.

In the practice of medicine maintaining and restoring homeostasis is a primary imperative. The clinical values of physiological variables that are considered homeostatic are referred to as 'normal values' (or normal ranges). These clinical normal values have a loose relationship to the cybernetic term 'set-point', which physiologists use to designate an imaginary target value that the system will return to after having been disturbed.

When the values for a regulated variable move outside the normal range, homeostasis has an increasingly difficult time, and medical interventions can be needed to avoid decompensation (breakdown of homeostasis). Values for a physiological variable therefore may be in the normal homeostatic range (near the set-point), homeostatically perturbed (outside normal values but recoverable through physiological reflexes) or decompensating (abnormal and requiring immediate intervention).

## Mechanisms distinguish kinorhesis from homeostasis

Kinorhesis differs from homeostasis not only because of different outcomes (metamorphosis is kinorhetic, and stable

blood sugar is homeostatic) but also because of the different mechanisms employed. These mechanistic differences can be understood generally from existing literature. However much more analysis of specific physiological instances will be necessary for a full understanding. The phenomenological categorization of physiological processes as either homeostatic or kinorhetic is an interesting exercise in itself, but the mechanistic differences have practical as well as theoretical importance.

The notion of 'mechanism' in a physiological system has two different meanings that are helpful here. The first version suggests that a mechanism can be an *a priori* construct that formally describes the components of the system and their relationships. Usually represented as a diagram the *a priori* mechanism is not anatomical or biochemical but rather represents information flow. A mechanism diagrammed in this *a priori* sense represents some version of the minimal elements that logic tells us are necessary to produce the outcome, which could be an autoregulatory negative feedback reflex.

The second version of mechanism is an empirical product consisting of anatomical and biochemical components that do the actual work described abstractly by an *a priori* mechanism. Experimental physiology is guided by *a priori* mechanism concepts to discover the anatomical and biochemical elements. An example of this empirical type of mechanism is the osmoregulatory negative feedback system (Carmody et al., 2015), for which a partial inventory of elements includes osmosensitive peptidergic neurons in the hypothalamus, secreted antidiuretic hormone (ADH), the bloodstream, kidneys, ADH receptors on collecting tubule cell membranes, cAMP, protein kinases and membrane vesicles harbouring aquaporin. Why is all this particularity important? The reason is not only that the particularity satisfies our curiosity but also that it provides information necessary to develop pharmaceutical or other medically useful interventions. Homeostasis and kinorhesis differ as to the abstract *a priori* mechanisms employed by each, and these distinctions guide the identification of empirical mechanisms.

The familiar mechanisms underlying homeostasis are commonly referred to as negative feedbacks. This terminology has been widely adopted across all of physiology for both research and teaching purposes. Despite this there are some ongoing discussions in the literature about whether negative feedback is exactly the correct formulation (McEwen & Wingfield, 2010; Ramsay and Woods, 2014; Bechtel & Bich, 2025). Operationally negative feedback seems like a fine phrase for many, but not all, types of regulatory mechanisms that are required for homeostasis, and much has been learned by applying this concept experimentally. Negative feedback systems are self-regulatory because the output of the system (e.g. secreted level of a hormone) is also the input to the feedback.

Certain terms (set-point, sensors, controller, etc.) are shared between physiology and engineering. It is understandable that sharing terms between physiology and engineering leads to some challenges or controversies. The set-point of a device is known because the operator chooses it, whereas a set-point in physiology can be inferred only from the behaviour of the system. Similarly components like sensors, controllers and effectors are concretely known in devices but must be discovered and inferred in physiology. Although negative feedbacks are conceptually similar in engineering and physiology, the heuristic values of feedback models are different. In engineering the model is a guide to designing and manufacturing useful regulatory devices. In physiology the regulatory device is pre-existing in nature, and the model is a guide to discovering what the device is made of and how it works.

Negative feedbacks, as a general type of regulatory mechanism, are common in physiological systems, but sometimes they are complex and convoluted in ways that make our general model of negative feedback seem inadequate (Ramsay & Woods, 2014; Bechtel & Bich, 2024; Bechtel & Bich, 2025). For example body temperature regulation in mammals cannot be reduced to a single negative feedback loop. Temperature regulation involves reflexes that control core temperatures in the visceral and cerebral circulation, and regional temperatures in skin and extremities. Processes controlled by these regulatory loops include everything from shivering, brown fat metabolism, regional vasoconstriction and vasodilatation, piloerection and ancient behavioural responses like seeking shelter and pulling on an extra blanket. Depending on one's perspective one might conclude that body temperature is either not controlled by negative feedback or controlled by interactions among multiple negative feedbacks that have evolved through many environmental challenges.

Even a very inclusive view of negative feedback leaves out other mechanisms that contribute to homeostasis. Some of these homeostatic mechanisms are as simple as kinetic properties of enzymes, transporters and solute channels that have been selected by evolution to match cellular and organismal needs with environmental conditions. An example of this from one's general physiology course is haemoglobin binding of oxygen. Oxygen binding to haemoglobin is co-operative such that the oxygen-rich lung environment promotes full saturation of haemoglobin, whereas the oxygen-poor tissues promote oxygen release. Additional features such as sensitivity to pH, $CO_2$ and some metabolic substrates tune the binding and unbinding properties to execute basic oxygen homeostasis very precisely (Boron & Boulpaep, 2016; Hall, 2021). Another type of homeostatic regulation that is not *per se* negative feedback is exemplified by phosphate homeostasis. In this case even though stable blood phosphate levels are essential, there are no 'phosphate sensors' that mediate responses to excursions of phosphate. Instead phosphate regulation is coupled metabolically to other substrates that are controlled by direct negative feedback mechanisms. Levels of calcium are the primary determinant of phosphate changes in the blood. This coupling occurs by direct chemical interactions between calcium and phosphate in blood, bone and cells, and by the actions of calcium-regulated hormones, including parathyroid hormone and calcitriol (Boron & Boulpaep, 2016; Hall, 2021). These examples illustrate that even though negative feedback is a powerful concept for understanding homeostasis, other stabilizing mechanisms have evolved to contribute to the stable internal milieu.

I focused on negative feedbacks in my papers (Horseman, 2025a, 2025b) to illustrate the rather obvious point that the stable internal milieu depends on stabilizing types of physiological mechanisms. Because negative feedbacks are a type of mechanism that is necessary for homeostasis, can we identify different types of mechanisms that are required for kinorhesis? I presented extended arguments and examples showing the involvement of positive feedback loops and sequential controls in kinorhetic

regulation (Horseman, 2025a, 2025b). Positive feedbacks are structurally similar to negative feedbacks, but rather than being self-inhibitory, the output variable induces itself, leading to ever-increasing output from the system. Positive feedbacks are naturally catastrophic (e.g. chemical and nuclear explosions). In physiology positive feedbacks are typically limited by the system reaching some end-state or breaking point that avoids catastrophic runaway. Delivery of the fetus(es) at parturition is a familiar end-state that breaks the positive feedback that drives labour (Gibb et al., 2006). Positive feedbacks are found across all of the kingdoms, and they typically function as disruptors of otherwise homeostatic systems, allowing the organisms to go through transformations into qualitatively different physiological, morphological and/or behavioural states (Horseman, 2025b).

Sequential control is another type of mechanism that drives kinorhetic transformations. For sequential control a physiological system is organized as a series of discretely different stages that follow one another in an orderly fashion. Products of a given stage bring about the next stage, with each transition being theoretically irreversible. The predictable trajectories of developmental changes are built upon sequential physiological control processes. These developmental changes generally rely on activations and repressions of genes. Metamorphosis and the ovarian cycle of egg maturation, ovulation and luteinization are examples of sequentially controlled kinorhetic processes. Sequential control processes in turn are controlled by signals from within and outside of the cells and tissues.

Positive feedback and sequential control mechanisms are evolved and evolvable physiological processes that drive kinorhetic changes so that development and reproduction can be accomplished. Positive feedbacks disrupt otherwise homeostatic systems, and sequential controls direct the qualitative changes that become manifest during development and reproduction.

### Does physiology need a second principle?

It is worth asking, is homeostasis sufficient by itself as the only governing principle for physiology? It is common to read that homeostasis is 'the' governing principle of physiology. If one logically followed the premise that homeostasis is sufficient to be the only organizing principle of physiology, then there are only two ways to think about the physiology of transformative life-history episodes. Either these processes must be excluded from physiology because they are not homeostatic, or the definition of homeostasis must be contorted so as to include events that are, by their very nature, the antithesis of stability (metamorphosis, pregnancy, etc.).

I have never read a direct argument for excluding reproduction and development from the scope of physiology (that would be strange). However the processes of reproduction or development are consistently absent from writings about homeostasis. And the tendency of physiologists to focus almost exclusively on homeostasis has meant that the elephant has sat unacknowledged in the room. As to contorting homeostasis so that it might include transformative processes, the only examples I have found that seem to make this argument resort to teleology, teleonomy and vitalism (Allen & Neal, 2020; Billman, 2020; Lennox & Kampourakis 2013; Turner, 2013; Turner, 2017; Vane-Wright & Corning, 2023). These are zombie ideas that keep getting resurrected and dressed up in supposedly new scientific garb. I should not need to argue again, as many others already have, that these sorts of reasoning may have their places but not in the performance of scientific learning.

To define a proper place for transformative physiology, this new theoretical treatment offers a general framework of explanations for developmental and reproductive processes while at the same time preserving the meaning of homeostasis. Kinorhesis addresses episodes of transformation that are required for driving development and reproduction during the course of a life history. Kinorhesis, like homeostasis, depends only on scientific mechanisms that can be studied experimentally, not on teleology and similar non-scientific approaches. Kinorhetic processes, like those of homeostasis, are subject to conventional evolutionary variability and selection. In fact for organisms to diversify and adapt to new or changing circumstances, the regulatory processes that underlie homeostasis and kinorhesis must evolve under selection.

Perhaps the most important point to emphasize again is that kinorhesis and homeostasis are not competing concepts. Homeostasis is continuous, and at least some homeostatic processes persist during kinorhetic transformations. Kinorhesis is episodic and drives particular life-history events. This is why I contend that these can be 'complementary principles of physiology'.

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

## Additional information

### Competing interests

None declared.

## Author contributions

All conceptualization and writing were performed by the only author.

## Funding

None.

## Acknowledgements

Thanks to John N. Lorenz for helpful discussions and suggested revisions to the manuscript.

## Keywords

development, feedback, life-cycle, reproduction, theory

## Supporting information

Additional supporting information can be found online in the Supporting Information section at the end of the HTML view of the article. Supporting information files available:

**Peer Review History**

