## [Peer Review History · The Journal of Physiology]

What Homeostasis Leaves Out: Kinorhesis, a Physiological Principle of Transformation

Nelson D. Horseman
DOI: 10.1113/JP291068

Corresponding author(s): Nelson Horseman (Horsemn@ucmail.uc.edu)

The following individual(s) involved in review of this submission have agreed to reveal their identity: Denis Noble (Referee #1); Janna L Morrison (Referee #2)

Review Timeline:

Submission Date:	29-Jan-2026
Editorial Decision:	22-Feb-2026
Revision Received:	25-Feb-2026
Accepted:	02-Mar-2026

Senior Editor: Kim Barrett

Reviewing Editor: Vaughan Macefield

Transaction Report:

Dear Dr Horseman,

Re: JP-OP-2026-291068 "What Homeostasis Leaves Out: Kinorhesis, a Physiological Principle of Transformation" by Nelson D. Horseman

Thank you for submitting your manuscript to The Journal of Physiology. It has been assessed by a Reviewing Editor and by 2 expert referees and we are pleased to tell you that it is acceptable for publication following satisfactory revision.

REVISION CHECKLIST:

We look forward to receiving your revised submission.

Yours sincerely,

Kim Barrett
Senior Editor
The Journal of Physiology

EDITOR COMMENTS

Reviewing Editor:

Thank you for submitting your revised manuscript to the Journal of Physiology. Both viewers are largely satisfied with your revisions, but have some additional small points that I would like you to attend to. I look forward to receiving your revised manuscript shortly.

REFEREE COMMENTS

Referee #1:

This article argues that, over and above homeostasis, and even homeorhesis, there is a principle of kinorhesis. This is the way in which physiology becomes directly relevant to evolutionary biology, as one of the examples in the article shows: fish to amphibian. Existing physiology must play a role in the process by which species evolve and this article provides a conceptual context for that process.

It may help for the author to consider referring to the special issue on physiology and evolution, published in 2024 in this journal. <https://doi.org/10.1113/JP284432>. That issue also argued for a primary role of existing physiology in the speciation process. George Romanes, a later collaborator with Charles Darwin, even coined the phrase Physiological Selection to describe the process. (volume 3 of Darwin and After Darwin 1886).

Referee #2:

Thank you for this interesting piece.

The concept is new and worth discussion. Homeostasis is a hallmark of physiology but I agree that there are many exceptions in reproduction and development. This is an interesting way to express these exceptions.

The piece is well written and engaging.

Line 224 - Should this be 'mechanisms'?

END OF COMMENTS

To the Editor:

We very much appreciate the reviewer comments. Two issues in need of changes were raised:

Reviewer 1. "consider referring to the special issue on physiology and evolution, published in 2024". This is an excellent suggestion. We had considered some of these issues in Horseman, 2025b and associating this with the J Phys special issue is very useful.

Reviewer 2. "Line 224 - Should this be 'mechanisms'?" The reviewer is correct, and I have made the change.

Thank you and the reviewers.
Nelson Horseman

Re: JP-OP-2026-291068R1 "What Homeostasis Leaves Out: Kinorhesis, a Physiological Principle of Transformation" by Nelson D. Horseman

Dear Professor Horseman,

We are pleased to tell you that your paper has been accepted for publication in The Journal of Physiology.

Yours sincerely,

Kim Barrett
Senior Editor
The Journal of Physiology

IMPORTANT POINTS TO NOTE FOLLOWING ACCEPTANCE OF YOUR PAPER:

- **IMPORTANT NOTICE ABOUT OPEN ACCESS:** To assist authors whose funding agencies mandate immediate public access to published research findings, The Journal of Physiology allows authors to pay an Open Access (OA) fee to have their papers made freely available immediately on publication.

The Corresponding Author will receive an email from Wiley with details on how to register or log in to Wiley Services where you will be able to place an order

- You can check if your funder or institution has a Wiley Open Access Account here: <https://authors.wiley.com/author-resources/Journal-Authors/open-access/author-compliance-tool.html>

- If you would like to receive our 'Research Roundup', a monthly newsletter highlighting the cutting-edge research published in The Physiological Society's family of journals (The Journal of Physiology, Experimental Physiology, Physiological Reports, The Journal of Nutritional Physiology, and The Journal of Precision Medicine: Health and Disease), please click this link, fill in your name and email address and select 'Research Roundup': <https://www.physoc.org/journals-and-media/membernews>

- You can help your research get the attention it deserves! Check out Wiley's free Promotion Guide for best-practice recommendations for promoting your work at: www.wileyauthors.com/eeo/guide. You can learn more about Wiley Editing Services which offers professional video, design, and writing services to create shareable video abstracts, infographics, conference posters, lay summaries, and research news stories for your research at: www.wileyauthors.com/eeo/promotion.

EDITOR COMMENTS

Reviewing Editor:

Thank you for including the additional references.